# Genetics and Neurobiology of Treatment-Resistant Depression—A Review

**DOI:** 10.3390/ijms262211016

**Published:** 2025-11-14

**Authors:** Olga Płaza, Piotr Gałecki, Katarzyna Bliźniewska-Kowalska, Małgorzata Gałecka, Agnieszka Brońska, Jan Płaza, Amelia Szurek, Agata Szulc

**Affiliations:** 1Department of Psychiatry, Faculty of Health Sciences, Medical University of Warsaw, 05-802 Pruszków, Poland; 2Department of Adult Psychiatry, Medical University of Łódź, 91-229 Łódź, Poland; 3Department of Psychotherapy, Medical University of Łódź, 91-229 Łódź, Poland; 4Students Scientific Association at the Department of Psychiatry, Faculty of Health Sciences, Medical University of Warsaw, 05-802 Pruszków, Poland

**Keywords:** treatment-resistant depression, major depressive disorder, pharmacogenomics, epigenetic modifications, neuroimaging, neuroanatomy, chronic inflammation, neurogenesis disfunction

## Abstract

Treatment-resistant depression (TRD), defined as the failure to achieve adequate response to at least two antidepressant trials, affects 20–30% of patients with major depressive disorder and poses substantial personal and socioeconomic burdens. This review aimed to synthesize current knowledge on the genetic, epigenetic, and neurobiological underpinnings of TRD to understand its pathophysiology better and inform future treatment strategies. A systematic search identified relevant studies focusing on genetic predispositions, epigenetic modifications, structural and functional brain alterations, the role of chronic inflammation, and deficits in neuroplasticity and neurogenesis associated with TRD. Findings highlight the involvement of polymorphisms in genes regulating neurotransmission, neuroplasticity, and stress response, though replication across studies remains inconsistent. Genome-wide association studies suggest polygenic contributions but are limited by small sample sizes and heterogeneous definitions of TRD. Emerging evidence points to aberrant DNA methylation, histone modifications, and dysregulated non-coding RNAs as potential mediators of treatment resistance. Neuroimaging studies reveal TRD-specific patterns, particularly altered default mode network connectivity and white matter disruptions, supporting its distinction as a subtype of depression. Collectively, the evidence underscores TRD as a multifactorial condition shaped by genetic and neurobiological factors, while emphasizing the need for standardized definitions, larger cohorts, and longitudinal designs to advance the field.

## 1. Introduction

Treatment-resistant depression (TRD) is a subtype of major depressive disorder (MDD), where no adequate response to treatment is observed, despite sufficiently long therapy with two different antidepressant medications in sufficient doses [1]. Despite continuing improvements in pharmacological and psychotherapeutic approaches, treatment-resistant depression affects approximately 20–30% of patients diagnosed with MDD [2]. This subtype of depression impacts not only patients and their loved ones, but is also a socioeconomic challenge, as it is related to increased risk of relapse, higher suicidality, and bigger socioeconomic costs [3].

The etiology of TRD is multifaceted, encompassing psychological, environmental, and biological factors. Identifying biological risk factors might allow healthcare professionals to earlier recognize patients at risk of treatment resistance, thus allowing for more personalized and quicker, more effective treatment.

This review aims to provide an analysis of potential biological risk factors of TRD. It focuses on genetic predispositions, epigenetic modifications, and neuroanatomical as well as functional brain alteration characteristics of TRD. Additionally, this paper pinpoints the limits in the current understanding of TRD etiology and shows potential future research directions, including identifying potential biomarkers for treatment resistance in patients with MDD.

## 2. Materials and Methods

Four electronic databases (PubMed, Embase, Cochrane Library and Web of Science) were searched for papers, with the first search focused on genetics and epigenetics of TRD, the second focused on neuroanatomy and neurofunctionality, and the third focused on inflammation and potential disruptions in neurogenesis and neural plasticity. In the first search, the keywords were as follows: “treatment-resistant depression” OR “treatment-refractory depression” and “gene” OR “GWAS” OR “pharmacogenomics” OR “candidate gene studies” OR “antidepressant resistance”. In the second search, the keywords “treatment-resistant depression” OR “treatment-refractory depression” and “neuroanatomy” OR “neurobiology” OR “MRI” OR “fMRI” were searched. In the third search the keywords “treatment-resistant depression” OR “treatment-refractory depression” and “inflammation” OR “neurogenesis” OR “neural plasticity” OR “synaptogenesis” OR “long-term potentation” were searched.

Of all the obtained results, a study was eligible for inclusion if the following criteria were met: an observational study design (clinical trial or randomized controlled trial) or a systematic analysis (meta-analysis or systematic review), the focus of the paper dedicated directly to either genetics or neurological anomalies (either anatomical or functional) or role of inflammation or impaired neuroplasticity and neurogenesis in treatment-resistant disorder, and the work itself published in a peer-reviewed journal.

The search strategy distinguished between observational studies (cohort, case–control, cross-sectional) and interventional trials (randomized controlled trials). Only studies involving adult human participants diagnosed with major depressive disorder according to DSM or ICD criteria, and meeting criteria for treatment-resistant depression (defined as failure to improve following two standardized course of treatment) and clearly reporting outcomes related to treatment response or resistance were included. Inclusion was limited to English and Polish due to availability of researchers—although it must be acknowledged that it might introduce selection bias to the papers analyzed. Papers published between January of 2000 and June of 2025 were included in the study.

Excluded were papers focused on depression in bipolar disorder, papers with no clear definition of treatment resistance, as well as animal and pediatric studies. Following the process, duplicates were excluded, as were the articles which did not meet the inclusion criteria or proved substantively insufficient throughout the full-text screening.

Additionally, reference lists of all relevant original and review articles were searched manually to identify additional eligible studies.

## 3. Results

### 3.1. Genetics of TRD

The clinical observation of MDD being a heterogeneous entity has its confirmation in genetic studies, with significant genetic differences observed among different MDD samples [4] and different subtypes of depression theorized to have different genetic backgrounds [5]. Whilst there is no doubt that the overall clinical picture also depends on patients’ environment, as well as nongenetic biological factors, both familial research and studies using more modern methods, such as genome-wide complex trait analysis, demonstrate the importance of the genetic component in antidepressant response [6]. More frequent familial presence, as well as certain characteristic clinical features (including higher suicide risk and higher comorbidity with anxiety) might suggest TRD to be a genetically separate subtype of MDD [7].

For this reason, identifying genetic variants related directly to treatment resistance (and not depression itself) might help create biomarkers of treatment response and, in the future, help in both earlier identification of patients at risk of developing TRD, as well as optimization of antidepressive treatment.

#### 3.1.1. Candidate Gene Studies

In studies dedicated to treatment resistance, the majority use the candidate gene approach, focusing on polymorphisms within genes involved in neurotransmission, neurotrophic signaling, and stress response. Although limited by hypothesis-driven designs and relatively small sample sizes, these studies have identified several genes of interest in terms of TRD risk factors. The results were combined in Table 1.

Genes involved in monoaminergic neurotransmission

The *SLC6A4* gene, encoding the 5-HTT serotonin transporter, has a functional polymorphism, with short (S) allele connected to lower transcription of the gene and poorer antidepressant response [8]. Additionally, in patients with the SS phenotype, smaller hippocampal volume was observed compared to carriers of the long (L) allele. In line with his results, the LL genotype is both less common in TRD patients [9,10] and is related to better treatment outcomes [8,11]. It is, however, important to note that meta-analyses dedicated to the subject yield mixed results, likely due to population stratification and environmental moderators [12].Multiple polymorphisms of the *HTR2A* gene, encoding the serotonin receptor 2A, have been studied regarding antidepressant response. The rs7997012 SNP has been linked with SSRI response in the STAR*D study [2], with AA homozygotes showing better response to antidepressant treatment in comparison to carriers of the G allele. Similarly, in clinical settings, the rs7333412 GG genotype has been reported to decrease response to antidepressants in comparison to A allele carriers [13]. Authors of meta-analyses dedicated to the subject emphasize, nevertheless, that although involvement of the gene is undeniably observed, it must be remembered that it is still a single gene; thus, its individual impact on treatment outcome is modest [14].*KCNK2* gene, encoding the two-pore domain potassium channel TREK1 involved in the serotonin transmission, was also analyzed as part of the STAR*D study continuation, with the rs2841608 allele identified as statistically significantly related to treatment resistance [15].Polymorphisms of the *COMT* gene, encoding catechol-O-Methyltransferase, an enzyme involved in catecholamine degradation, have been related to increased suicidality [16]. Carriers of the A allele rs4680 SNP (either mutant G/A heterozygotes or mutant A/A homozygotes) reported TRD significantly more often, with the heterozygous mutant genotype showing the worst response to ECT [17].Carriers of the C variant of the rs2242446 SNP of the *SLC6A2* gene, which codes the norepinephrine transporter, were found to have smaller hippocampal volume compared with noncarriers [18]. This polymorphism was also observed more frequently in TRD patients in comparison to healthy controls—one must, however, note the small sample size and lack of replication of the result so far [10].

Genes involved in neuroplasticity and stress response

The *GRIN2B* gene, encoding the subunit of the NMDA receptor, has been researched as a potential candidate gene for MDD and TRD susceptibility. The rs1805502 polymorphism (dominance of G allele) has been linked to TRD presence [19].Multiple alleles of the *GRIK4* gene (encoding the Glutamate Receptor Ionotropic Kainate 4) have been analyzed, with results pointing to the gene’s involvement in the pathogenesis of TRD. The G allele of the rs11218030 SNP was observed to positively correlate with TRD presence. Additionally, rs1954787 GG homozygotes and G allele carriers of rs11218030 have a higher risk of developing psychotic symptoms during a depressive episode [20] and worse response to electroconvulsive therapy.In patients with TDR, lower serum levels of *BDNF* are observed. This can be explained by the role of rs6265 polymorphism—presence of the Met allele is correlated with a lower protein concentration and has been correlated with TRD in preclinical studies [21] and a pilot study in a mainly European population [22] (it must however be noted, that these results were not replicated in a study focused on Chinese participants, thus emphasizing the importance of analyzing general genetics of population beforehand). In a different study, Met allele of the SNP was correlated with a worse response to rTMS treatment [11].The *NR3C1* gene encodes the glucocorticoid receptor involved in HPA axis regulation. Presence of the rs6198 and rs41423247 SNPs has been explored concerning antidepressant resistance.

**Table 1 ijms-26-11016-t001:** Genes analyzed through candidate gene studies, with potential relation to TRD.

**Genes Involved in Monoaminergic Neurotransmission**				
**Gene**	**Polymorphism**	**Sample Analyzed**	**Results**	**Source**
*SLC6A4*	rs25531 (S allele)	310 patients with TRD and 284 healthy controls	Increased risk of treatment resistance	[8]
	rs25531 (LL phenotype)	310 patients with TRD and 284 healthy controls	Smaller risk of treatment resistance	
*HTR2A*	rs7997012 (G allele)	3671 patients with MDD	Increased risk of treatment resistance	[2]
*KCNK2*	rs2841608 (C allele)	1554 patients with MDD	Increased risk of treatment resistance	[15]
*COMT*	rs4680 (A allele)	100 patients with TRD (of Chinese descent) and 100 healthy controls	Increased risk of treatment resistance Heterozygous genotype (G/A) showing the worst response to ECT	[17]
*SLC6A2*	rs2242446 (C allele)	26 TRD patients and 27 matched healthy controls	Smaller hippocampal volume	[18]
		119 TRD patients (of Finnish descent) and 395 healthy controls	Increased risk of treatment resistance	[10]
**Genes Involved in Neuroplasticity and Stress Response**				
**Gene**	**Polymorphism**	**Sample Analyzed**	**Results**	**Source**
*GRIN2B*	rs1805502 (G allele)	178 TRD and 612 non-TRD patients as well as 779 healthy controls	Increased risk of treatment resistance	[19]
*GRIK4*	rs1954787 (GG homozygotes)	247 MDD no-TRD and 380 TRD patients	Higher risk of developing psychotic symptoms during a depressive episode Worse response to electroconvulsive therapy	[20]
	rs11218030 (G allele)		Increased risk of treatment resistance Higher risk of developing psychotic symptoms during a depressive episode Worse response to electroconvulsive therapy	
*BDNF*	rs6265 (Met allele)	31 patients with MDD, defined as treatment-resistant MDD defined using DSM-IV criteria	Worse response to rTMS treatment	[11]
		62 patients (55% female patients) of mainly European ancestry (58 Patients) with MDD	Carriers of Met allele show lower response to ketamine treatment	[22]
*NR3C1*	rs6198 and rs41423247 alleles	760 patients with moderate-to-severe depression, treated with escitalopram or nortriptyline	Alleles correlated with worse response to treatment with SSRI (escitalopram) and nortriptyline (tricyclic antidepressant)	[21]

#### 3.1.2. GWAS

Although early GWAS for MDD showed few replicable findings due to limited sample sizes, more recent efforts—particularly those from large consortia such as the Psychiatric Genomics Consortium (PGC)—have identified numerous loci associated with depression [23].

Tansey et al. [6] conducted one of the first large-scale GWAS of antidepressant response using data from the GENDEP and STAR*D cohorts. While no SNPs reached genome-wide significance for treatment resistance per se, the study provided a polygenic score model that explained a small proportion of variance in treatment outcomes. Similar results were obtained by Li et al.—no genome-wide significant or suggestive signal was identified. The study, however, allowed for identification of alleles of the *GPRIN3* gene related to bupropion response [24] (each copy of the rs1908557-C allele was associated with higher odds of being a bupropion non-responder).

Emerging evidence also highlights the contribution of inflammation-related genetic variants to treatment resistance and antidepressant response. For example, Draganov et al., in a study focused solely on variants and methylation status of genes related to inflammation, identified an SNP of *IL1-β* (rs1143643) correlated with treatment resistance measured via the Maudsley Staging Method (MSM) [25]—a result in line with previous studies associating this allele with failure to respond to SSRI treatment [26]. Another SNP of this gene (rs16944) has also been correlated with worse response to antidepressant treatment, particularly when correlated with adverse life history [26,27,28]. Interestingly, carriers of this allele also showed reduced activation of the amygdala and anterior cingulate cortex (ACC) in response to emotional stimuli, suggesting a neural mechanism through which the genotype might influence treatment response [26]. An allele of the *IL6* gene promoter (rs1800795) has also been associated with worse outcomes, including treatment resistance [29]. The studies, however, vary in reports of impact severity, although there seems to be a consensus regarding this polymorphism to interact with environmental stressors and painful physical conditions to influence depressive symptom severity (though not always treatment outcome per se) in population cohorts [25,30]. For *TNF*, the rs1800629 variant has been tested in several case–control and meta-analysis designs, but direct evidence linking it to treatment resistance is limited. A meta-analysis on the matter found no statistically significant association between rs1800629 and overall depression susceptibility and treatment response [31]. In case of C-reactive protein (CRP), majority of studies focus on serum concentration and its potential impact on pathophysiology of depression, with scarce studies focused on potential SNPs impact on serum levels and not antidepressant response or remission vs. resistance [32].

In a study focused on copy number variants, TRD was associated with an increase in duplications, as well as a deletion in the *PABPC4L* gene (encoding a protein involved in the degradation of abnormal mRNA). The results, however, were not obtained again following multiple-testing correction [33].

In all GWAS studies, researchers point out the challenge presented by methodological limitations, including small sample sizes (resulting in insufficient statistical power), small effect sizes of common risk loci associated with TRD, and the relatively limited coverage of common human genetic variation [7].

A potential solution is to avoid the necessity of recruiting large samples, the use of aggregated tests and genotype imputation using large and diverse reference panels, and machine learning models for analysis. This allows for GWAS analysis focused on pathways, such as one carried out by Fabri et al. [34], which allowed for the identification of alleles of the *CACNA1C* gene (encoding a calcium channel) directly correlated with TRD presence. The same study allowed for identification of other genes in the identified pathway that are linked to long-term potentiation, neural survival, neurogenesis, and neuroplasticity, but also to MDD and antidepressant efficacy. Among these genes, some regulated glutamate receptors (e.g., *TRPM4*, *PIK3CG*, *SUMO1*) or calcium currents modulating the process of long-term potentiation (e.g., *CACNA1C*, *CAMK2D*, *FKBP1B*, *P2RX4*, *RYR2*) were involved. It is interesting to note that a previous GWAS identified a gene set involved in transporter activity as a possible modulator of antidepressant response in two samples [35]. This gene set includes *CACNA1C* and other calcium channel coding genes such as *CACNB2*; the latter gene was also identified as involved in the shared genetic susceptibility to several psychiatric traits including MDD [23].

It is also important to note the lack of standardization of TRD phenotype (with many studies relying on qualities which must not necessarily reflect actual treatment resistance, such as the number of hospital admissions), and the fact that not many large prospective pharmacogenetic trials stratify by genotype and follow treatment resistance endpoints beyond 4–6 weeks and most do not consider environment interactions and/or epigenetic regulation further impact the research on the matter.

### 3.2. Epigenetic Modifications in TRD

Epigenetics refers to heritable changes in gene expression that do not involve alterations in the DNA sequence, with principal epigenetic mechanisms being DNA methylation, histone modifications, and non-coding RNAs. These modifications are influenced by environmental, developmental, and pharmacological factors and they regulate gene activity dynamically, thus influencing the expression of genes involved in neuroplasticity, neurotransmission, and immune function. Epigenetic plasticity also offers a mechanistic explanation for the delayed onset of antidepressant effects. Chronic treatment may be required to reprogram maladaptive gene expression patterns. Furthermore, non-pharmacological interventions such as psychotherapy, mindfulness, and exercise have been shown to modify epigenetic marks, potentially enhancing their therapeutic efficacy in TRD [36].

The majority of research regarding MDD and TRD and the potential role of epigenetics in their pathophysiology is still in the very early stages, with most dedicated to animal models or small samples of human patients. Thus, all the results described below have to be analyzed taking that into account.

#### 3.2.1. DNA Methylation

Numerous studies have implicated aberrant DNA methylation in MDD and TRD. For example, increased methylation of the *BDNF* promoter has been observed in individuals with depression and correlates with reduced BDNF expression in post-mortem brain tissue [37]. In patients with TRD, hypermethylation at BDNF loci may contribute to impaired neuroplasticity and poor treatment response.

Another frequently studied gene is *NR3C1*, which encodes the glucocorticoid receptor. Hypermethylation of NR3C1 has been linked to altered stress reactivity and reduced feedback inhibition of the hypothalamic–pituitary–adrenal (HPA) axis—a hallmark of TRD [38]. These epigenetic alterations often reflect early life adversity, suggesting a critical role for developmental epigenetics in TRD vulnerability.

Genome-wide methylation analyses have also identified novel loci associated with antidepressant response. In a large-scale study by Klengel et al. [39], methylation changes in *FKBP5*—a gene modulating glucocorticoid receptor sensitivity—were found to interact with childhood trauma in predicting antidepressant non-response. Similarly, increased methylation of the SLC6A4 gene (encoding the serotonin transporter) has been associated with poorer SSRI response [40].

Adverse childhood experiences (ACEs) are robust predictors of TRD and have been shown to alter the methylation of genes involved in HPA axis regulation, immune response, and neurodevelopment [41]. For instance, methylation of the *OXTR* gene, which encodes the oxytocin receptor, has been linked to emotional regulation and social functioning in depression.

#### 3.2.2. Histone Modifications

Preclinical models of depression have shown that stress and antidepressant treatment alter histone acetylation patterns in brain regions such as the hippocampus and prefrontal cortex [42]. Tsankova et al. [43] demonstrated that chronic social defeat stress in mice reduced histone acetylation at the BDNF promoter (what is interesting, chronic imipramine treatment showed a potential reversal of these effects).

In TRD, dysregulated histone deacetylase (HDAC) activity may hinder the transcription of genes required for neurogenesis and synaptic plasticity. Inhibitors of HDACs have shown antidepressant-like effects in animal models, and pilot studies suggest their potential as adjunctive treatments in TRD [44].

Additionally, alterations in histone methylation—both activating (e.g., H3K4me3) and repressive (e.g., H3K27me3)—have been linked to stress resilience and susceptibility. The histone methyltransferase G9a, for instance, regulates gene repression in response to stress and has been implicated in chronic antidepressant efficacy [45].

#### 3.2.3. Non-Coding RNAs: miRNAs and lncRNAs

Non-coding RNAs, including microRNAs (miRNAs) and long non-coding RNAs (lncRNAs), play a crucial role in post-transcriptional gene regulation. miRNAs function by binding to complementary sequences in target mRNAs, leading to degradation or translational repression.

Several miRNAs have been implicated in the pathophysiology of MDD and TRD. For example, miR-1202, which regulates the glutamate receptor GRM4, is downregulated in depressed patients and normalizes with antidepressant treatment [46]. Reduced levels of miR-135a, a regulator of serotonin signaling, have also been associated with treatment resistance [47]. miR-146a and miR-155, which regulate inflammatory pathways, may distinguish responders from non-responders to antidepressants [48].

The utility of circulating miRNAs as peripheral biomarkers is particularly promising. Changes in miRNA profiles in blood or saliva may reflect central nervous system alterations and could aid in predicting treatment outcomes or monitoring therapeutic response in TRD patients.

### 3.3. Neuroanatomical Brain Changes

Improvement in neuroimaging techniques allowed for the identification of structural and functional brain abnormalities associated with major depressive disorder. Those including hippocampal volume reduction were most pronounced in the cornu ammonis, dentate gyrus, and subiculum [49,50] as well as frontal atrophy, particularly in the medial prefrontal cortex, frontal cortex and dorsolateral prefrontal cortex [51,52] and decreased volume of gray matter nuclei [53]. Although all this can be observed in patients with TRD, some other specific patterns of neuroanatomical changes have been identified as characteristic of treatment-resistant depression. Therefore, to focus on TRD-specific changes, all studies analyzed below include not only patients with TRD and healthy controls, but also a third group of patients with history of MDD in general, who did not meet the criteria of treatment-refractory depression. The results are combined in Table 2.

#### 3.3.1. Gray Matter Volume

When comparing gray matter volume (GMV) in patients with TRD, compared to healthy controls and patients in first-episode MDD, Ma et al. [54] observed TRD-specific reduction in caudate volume, which was related to altered caudate–prefrontal connectivity, suggesting a potential cause of dysregulation of the reward mechanism. Shah et al. [55] compared TRD patients to patients who recovered from recurrent MDD and healthy controls and observed reduced gray matter density in the right superior frontal gyrus and the right putamen in the TRD group, as well as changes in tissue composition in the hippocampus and rostral ACC. Additionally, the results describe a statistically significant reduction in volume of the right prefrontal lobe and the right caudate nucleus in the TRD group. Similar results were observed by Serra-Blasco et al. [56]—compared to healthy controls, TRD patients showed reduced GMV in superior, medial and inferior frontal gyri, the insula, the parahippocampal gyrus, transverse temporal gyrus and anterior cingulate gyrus; in comparison to the first-episode group, the TRD group revealed smaller volumes of the right medial frontal gyrus and the left insula in comparison with those in the first-episode group, as well as qualitative GMV changes compared with the first-episode group in the precentral gyrus, the medial frontal gyrus, the insula, the transverse temporal gyrus, the inferior parietal lobule, and the posterior cingulate.

#### 3.3.2. Regions of Interest

Studies focused on hippocampal anatomy showed that when comparing patients with different stages of MDD, those with TRD showed reduced volume of the tail section of the hippocampus, as well as significantly more sulcal cavities [57,58].

#### 3.3.3. White Matter Structure

When analyzing white matter microstructure in healthy controls and patients in different stages of MDD (treatment-resistant, remitted–recurrent and first-episode), significant reductions in fractional anisotropy in the cingulum, corpus callosum, superior and inferior longitudinal fasciculi were observed in TRD patients compared to healthy controls and first-episode patients. When compared with MDD patients responsive to treatment, decreased fractional anisotropy was observed within the ventromedial prefrontal region in treatment-resistant patients [59]. This suggests that disruptions of white matter microstructure, particularly in fronto-limbic networks, are associated with resistance to treatment.

**Table 2 ijms-26-11016-t002:** Neuroanatomical brain changes related to TRD.

**Gray Matter Volume (GMV)**		
**Neuroanatomical Variant**	**Sample Analyzed**	**Source**
Reduced caudate volume, related to altered caudate–prefrontal connectivity	18 TRD patients and 17 patients with first episode of depression	[54]
Reduced gray matter density in the right superior frontal gyrus and the right putamen, changes in tissue composition in the hippocampus and rostral ACC, reduction in volume of the right prefrontal lobe and the right caudate nucleus	20 TRD patients, 20 patients with MDD who responded to treatment, 20 healthy controls.	[55]
Reduced GMV in superior, medial and inferior frontal gyri, the insula, the parahippocampal gyrus, transverse temporal gyrus, and anterior cingulate gyrus Smaller volumes of the right medial frontal gyrus and the left insula, qualitative GMV changes in the precentral gyrus, the medial frontal gyrus, the insula, the transverse temporal gyrus, the inferior parietal lobule and the posterior cingulate (in comparison to first-episode patients)	22 TRD patients, 22 patients with MDD who responded to treatment, 20 patients with first episode of depression	[56]
**Regions of Interest**		
**Neuroanatomical Variant**	**Sample Analyzed**	**Source**
Reduced hippocampal volume	182 TRD patients, 52 patients with schizophrenia, 76 healthy controls	[50]
Increased presence of hippocampal sulcal cavities	115 TRD patients, 86 healthy controls	[51]
**White Matter Structure**		
**Neuroanatomical Variant**	**Sample Analyzed**	**Source**
Reductions in fractional anisotropy in the cingulum, corpus callosum, superior, and inferior longitudinal fasciculi (in comparison to healthy controls and first-episode patients) Decreased fractional anisotropy within the ventromedial prefrontal region (in comparison to MDD patients responsive to treatment)	18 TRD patients, 16 patients with MDD who responded to treatment, 19 patients with first episode of depression	[59]

### 3.4. Functional Brain Changes

#### 3.4.1. Default Mode Network

One of the main areas of research regarding the functional aspect of neurobiology in treatment-resistant depression, is research dedicated to the default mode network (DMN)—the network active during self-referential thought and “mind-wandering”, primarily composed of the dorsal medial prefrontal cortex, posterior cingulate cortex, precuneus and angular gyrus. In research dedicated to functional connectivity within the network, majority of studies report a general decrease in it. In a previous study, Ma et al. [54] found significantly decreased connectivity between the right caudate and the right middle frontal gyrus and right superior frontal gyrus in treatment-resistant depression, whereas He et al. [60] report a significant reduction in connectivity, particularly regarding the connections between parahippocampal gyrus and the left precuneus, left posterior cingulate gyrus, and the left inferior parietal lobe which in turn had a reduced connection with the right caudate. Additionally, decreased connectivity between the DMN and other functional networks and brain regions was found in TRD compared to control groups. De Kwaasteniet et al. [61] reported a decreased functional connectivity between the left and right angular gyrus (the posterior DMN) and the cognitive control network, as well as a decrease in functional connectivity between both the anterior and posterior DMN and the visual cortex. What is more, they further found that in patients with treatment-resistant depression, there was significantly decreased functional connectivity between the motor cortex and other brain regions, therefore suggesting that in this group of patients, reduced resting-state connectivity is widespread and not just between specific neurocognitive networks. However, in studies focused on regional spontaneous neuronal activity in the DMN, contrary to results reported above, majority of studies reported an increase in spontaneous neuronal activity within its components. Increased local activity (measured using the fractional amplitude of low-frequency fluctuations—(f)ALFF-technique) was reported in the anterior cingulate cortex and medial frontal gyrus [62], as well as right thalamus and the supramarginal gyrus, at the edge of the angular gyrus [63]. In a study using a different technique (regional homogeneity or ReHo analysis—which measures local synchronization of rs-fMRI signals), a significant increase in activity in the right middle temporal and the middle cingulate gyri was reported in TRD patients [64].

#### 3.4.2. Occipital Region Connectivity Disfunction

Interestingly, occipital region of the brain shows impaired functionality and/or connectivity in patients with TRD, with research describing decreased functional connectivity between the anterior DMN and the cuneus [61], as well as between the olfactory cortex and the left and right inferior occipital gyrus and the left fusiform gyrus [60]. Additionally, Guo et al., in partially overlapping samples, described decreased interhemispheric functional connectivity in TRD in the occipital part of the fusiform gyrus and the calcarine cortex, as well as substantial decreased functional dysfunction with other regions in the occipital lobe and with the right inferior temporal gyrus and right insula in TRD patients [62,65]. The same group of patients also showed significantly increased coherence-based ReHo in the occipital part of the left fusiform gyrus and significantly decreased ALFF values in the lingual gyrus/cuneus, respectively. Those results were replicated in a different study, where decreased ALFF values in the right lingual region and increased fALFF in the middle occipital gyrus in TRD patients compared to non-TRD patients [63] were found. Summary of all findings regarding functional brain changes in TRD are summarised in Table 3.

### 3.5. Neurogenesis and Neural Plasticity Disfunction

Neural plasticity encompasses a broad set of mechanisms including synaptogenesis, dendritic remodeling, and long-term potentiation, all of which allow the brain to adapt both structurally as well as and functionally to environmental stimuli. In adult patients, brain plasticity is mainly related to new neurons deriving from neural progenitor cells in the subgranular zone of the hippocampal dentate gyrus and subventricular zone [49,50]. Increasing evidence suggests that impaired neurogenesis and defective neuroplasticity play a central role in the pathophysiology and persistence of TRD.

#### 3.5.1. Structural Correlates

Structural neuroimaging consistently shows reduced hippocampal volume in TRD, particularly in neurogenic subfields such as the dentate gyrus and CA3 region [49,66]. Post-mortem studies further confirm decreased density of neural progenitor cells and reduced expression of neurogenic markers such as doublecortin (DCX) and Ki-67 in hippocampal tissue from depressed individuals [67]. These findings suggest that diminished hippocampal neurogenesis may underlie cognitive rigidity and emotional dysregulation observed in chronic and treatment-refractory forms of depression.

In parallel, alterations in white matter tracts and fronto-limbic network connectivity have been reported in TRD, consistent with a generalized failure of neural plasticity [61]. Disrupted functional connectivity between the hippocampus, anterior cingulate cortex, and prefrontal regions impairs top–down regulation of affect and may contribute to the persistence of maladaptive emotional processing.

#### 3.5.2. Cellular and Molecular Mechanisms

Neuroplasticity depends on a coordinated interplay of neurotrophic factors, glutamatergic signaling, and intracellular pathways controlling synaptic remodeling. Among these, brain-derived neurotrophic factor (BDNF) and vascular endothelial growth factor (VEGF) are key regulators of neuronal survival and synaptogenesis. Reduced circulating and cortical BDNF levels are consistently observed in TRD [68,69], accompanied by decreased TrkB receptor activation. Additionally, lower VEGF expression has also been associated with poorer antidepressant response [70].

Chronic stress activates the HPA axis, resulting in elevated glucocorticoid exposure that inhibits neural progenitor proliferation and dendritic growth [71,72]. Moreover, persistent inflammation—marked by upregulation of IL-1β, TNF-α, and IL-6—impairs hippocampal neurogenesis by suppressing BDNF transcription and promoting microglial activation [73,74,75]. This inflammatory suppression of neurogenesis is further amplified by activation of the kynurenine pathway, leading to accumulation of neurotoxic metabolites that interfere with glutamatergic neurotransmission [76]. This further emphasizes the role of chronic inflammation and stress in patophysiology for treatment-resistant depression. Animal models show that chronic stress suppresses both hippocampal neurogenesis and synaptic protein expression, effects that persist despite traditional monoaminergic antidepressant treatment [77].

At the synaptic level, abnormalities in glutamate receptor signaling play a crucial role, with dysregulation of NMDA receptor subunits (GRIN1 and GRIN2B) and deficits in AMPA receptor trafficking reducing long-term potentiation and dendritic spine formation [78]. In TRD, such synaptic rigidity may prevent antidepressant-induced reorganization of cortical and limbic circuits.

Taken together, these findings reinforce the view that TRD is a disorder of “neuroplastic insufficiency.” Rather than reflecting mere neurotransmitter imbalance, TRD involves structural and molecular barriers to neural remodeling.

## 4. Discussion

In terms of genetic background, there is a scarcity of studies dedicated to just treatment-resistant depression compared to research focused on general antidepressant effectiveness. The most replicable results were obtained by candidate gene studies, however, whilst it allows for more detailed research, it also limits the number of genes which are being analyzed, as genes related to antidepressant response do not have to be directly correlated to their mechanism of action and focusing research to only those which are limits the amount of polymorphisms studied. The GWAS dedicated to TRD is singular, with no study identifying a genome-wide significant signal at a single variant level. This might partially be related to the fact that while GWAS analysis seems like a perfect method for studying pathologies of polygenic background, it requires big samples for analysis. Studies carried out so far focused on a rather small number of known variants, therefore they had a mostly inadequate power to detect signals with small effect size. For scale, approximately 40 million common variants were identified in the human genome using whole-genome sequence data from a number of cohorts, with previous GWAS of antidepressant response focused on the number of common variants between 1 and 9 million (so less than 25% of all potentially available). Additionally, small sample sizes present another difficulty—in differentiating actual small effects from false positives. It must, however, be noted that similar results were obtained in GWAS research dedicated to general antidepressant response—no genome-wide significant polymorphisms has been identified so far.

Two main ways of improving the quality of genetic research dedicated to the matter are discussed by specialists—firstly improving the coverage of genetic variants, and secondly, improving and modifying methods of statistical analysis. In terms of enlarging the amount of data available, all the consortia point out the method of sequencing and its optimization with imputation (a method of statistical analysis, allowing for inference regarding unobserved genotypes following tests of association between known haplotypes, traits of interests and experimentally untyped genetic variants [79]; it proved efficient in European and Asian populations). Imputation allows for good coverage of common genetic variants, combining it with sequencing would allow for researching the role of rare variants. This seems particularly promising taking into consideration growing availability of sequencing as a technique and methodological improvement.

Optimizing methods of genetic testing would also allow for developing more research into epigenetic modifications related to treatment-resistant depression. It is an established fact that environmental stressors and genetic factors shape the complex phenotype of MDD [42]—in a more precise way, epigenetic factors also impact the subtype of MDD which is TRD. As of right now, research dedicated to epigenetics of major depressive disorder is still a new branch of medicine, with data regarding DNA methylation, histone modifications, and non-coding RNA and their impact on pathophysiology still emerging. Continuous analyses are necessary, which would, in the future, allow for potential development of biomarkers of disease and treatment response, which could prove useful in understanding the mechanisms behind TRD development.

In terms of neuroanatomical and neurofunctional risk factors, results obtained so far seem to prove the hypothesis of TRD being a separate clinical entity, and a genuine subtype of MDD, with the most distinct differentiating feature being alterations in the DMN [80]. Although there are some neuroanatomical features more common in brains of patients with TRD, no results are suggesting them to be TRD-specific, thus suggesting the more important role of functional aspects in the pathogenesis. Researchers point to the limited number of studies dedicated to the matter of neuroanatomical and neurofunctional changes, with most of them focused on resting-state fMRI and structural MRI, which limits the data potential. Additionally, many studies use partially overlapping samples, which further limits the application of results. Thus it seems necessary for more independent research to be conducted, with larger groups of patients analyzed, for any result to be determined. Other potential improvement in future studies would be designing more prospective longitudinal studies (in which patients would be assessed after each treatment attempt, which would allow for following the dynamics of potential changes in both neuroanatomical and neurofunctional scans).

An underexplored area in understanding TRD as an entity is the role of inflammation and immune dysregulation in both its patophysiology and treatment sensibility. Multiple studies have shown elevated levels of pro-inflammatory cytokines such as interleukin (IL)-1β, IL-6, tumor necrosis factor-α (TNF-α), and C-reactive protein (CRP) in patients with depression [74,81,82], particularly in those with treatment-resistant depression [83]. This allows for the suggestion that chronic systemic inflammation might play a role in the resistance to traditional means of antidepressive treatment, as elevated cytokine levels are known to disrupt monoamine synthesis, instead increasing neurotoxic kynurenine pathway activity [76]. Additionally, inflammatory proteins increase glucocorticoid receptor resistance and impair feedback inhibition within the HPA axis, thus leading to a continuous state of hypercortisolemia that is linked to hippocampal atrophy and emotional dysregulation [84,85,86]. What is more, neural stem cells residing in the subventricular and subgranular zones show reduced proliferative capacity under chronic stress and inflammation [71].

Emerging findings seem to further prove the crucial role of immune pathways in TRD—studies mentioned above allowed for identification of SNPs associated with antidepressant resistance [25], which also seem to, at the same time, be playing a role in modulating neural functionality in emotion-processing areas of the brain [26]. Inflammation has a profound impact on neuroplasticity, as pro-inflammatory cytokines inhibit the expression of BDNF [87], therefore impairing synaptic remodeling, dendritic spine growth, and long-term potentiation. Neuroimaging evidence demonstrates that peripheral inflammation correlates with decreased reward circuit activity [88], suggesting that systemic immune activation directly relates to changes in neural network dynamics. What is more, pro-inflamamtory proteins inhibit progenitor survival and differentiation [73], therefore suggesting the role of chronic inflammation in decreasing the brain’s regenerative potential.

Beyond the role of genetic and inflammatory mechanisms, growing evidence suggests that disturbances in neurogenesis and neural plasticity are fundamental to the pathophysiology of TRD. Chronic stress, excessive glucocorticoid exposure, and neuro-inflammatory cytokines suppress progenitor proliferation and dendritic complexity, particularly in the hippocampus, a key region for emotion regulation, memory consolidation, and cognitive flexibility, thus further impacting the brain’s capacity for adaptive plasticity and its sensitivity to “classic” antidepressant treatment. SSRIs and tricyclic antidepressants promote hippocampal neurogenesis and synaptic plasticity [89,90]; in an experimental setting, ablation of hippocampal neurogenesis prevented behavioral recovery following antidepressant treatment, thus providing another proof of the importance of neurogenesis on therapeutic efficacy in antidepressive treatment [91]. In TRD, however, persistent stress exposure, increased inflammation and epigenetic repression of neurotrophic genes may blunt or prevent these neuroadaptive responses, resulting in and poor treatment outcomes [37,74].

An integrative perspective combining genetic, epigenetic, and neurobiological evidence suggests that TRD emerges from the interaction between inherited susceptibility (including inflammatory gene polymorphisms), environmentally mediated epigenetic changes, neural network dysregulation (notably within the default mode network and hippocampal circuitry), and defects in neurogenesis and neural plasticity. This system-level approach emphasizes cross-talk between immune, stress, and neuroplasticity pathways in shaping treatment resistance. Multidimensional biological background of TRD is summarised in Figure 1.

Interestingly, this hypothesis might also explain the high efficency of physical and neuromodulatory therapies, which appear to at least partially reverse biological deficit characteristics for depression, and TRD in particular. Electroconvulsive therapy (ECT) increases GMV, particularly in the hippocampal region, enhances BDNF and VEGF expression, and lowers circulating cytokines in patients with depression [92,93,94]. Similar results were obtained following anaylsis of patients who underwent rTMS and tDCS, with elevation in BDNF levels as well as promotion of cortical plasticity and improvement in fronto-limbic connectivity observed [95]. Rapid-acting antidepressants such as ketamine bypass monoaminergic mechanisms by activating the mTOR and BDNF-TrkB pathways, promoting synaptogenesis and dendritic spine formation within hours of administration [96,97]. Psychedelic-assisted therapies, including psilocybin, may act through 5-HT2A receptor-mediated neuroplastic mechanisms that facilitate cognitive and emotional flexibility, also offering promise for TRD [98,99]. Additional interventions such as vagus nerve stimulation and deep brain stimulation (DBS) modulate neurotrophic and immune pathways, showing promising additional treatment strategies [100,101]. Methods of modifying signaling pathways (including agents that modulate Wnt/β-catenin, Notch, or Sonic Hedgehog) might also help restore neural stem cells and thus improve neurogenesis [102]. This might allow for development of stem-cell-derived exosome therapies, rich in neurotrophic and anti-inflammatory factors, by mimicking the effects of endogenous repair mechanisms.

Collectively, results related to new therapeutic methods seem to suggest that successful antidepressant interventions share a common endpoint-restoration of neuroplastic capacity and neural resilience, with failure of this process correlated with treatment resistance. Hence, therapeutic strategies that integrate neurotrophic enhancement, anti-inflammatory modulation, and environmental enrichment may represent the most effective approaches for promoting long-term recovery in TRD.

The consensus among researchers is the need for more unified criteria of treatment-resistant depression, as well as more precise identification of other factors which might impact patients’ state (including socioeconomic situation, as well as presence of other treatment methods such as psychotherapy). Whilst in most of the research published recently there seems to be an agreement regarding the definition of TRD (failure of two or more antidepressant treatments), there is often no information regarding the total number of pharmacotherapy attempts, illness duration, severity of depression symptoms in each episode, as well as no information regarding potential psychosocial and socioeconomic differences between patients. Lack of such information not only makes it impossible to identify patients with different levels of treatment resistance (all of whom might present with their own neurobiological and genetic characteristics), but also limits the potential of developing effective interventions for TRD. Future research should integrate genetic, epigenetic, and neurobiological dimensions using larger, well-characterized cohorts. Multimodal analyses combining genomic sequencing with imputation, inflammatory profiling, and neuroimaging biomarkers may yield a more comprehensive understanding of TRD pathophysiology and enable the identification of mechanistically grounded therapeutic targets.

## 5. Limitations

This review was focused on biological and molecular determinants of treatment-resistant depression, thus it intentionally does not cover psychosocial and behavioral determinants known to influence illness course and treatment outcomes—such as early life trauma, chronic stress, low socioeconomic status, social isolation, and barriers to healthcare access.

Recent studies demonstrate that socioeconomic disadvantage is associated with poorer treatment outcomes, suggesting that social context may contribute to the persistence of treatment resistance [103,104,105]. Similarly, research published in recent years indicate that high intake of simple sugars and ultra-processed foods correlates with greater depressive severity and lower treatment responsiveness [106,107], which seems to be related to diet-mediated inflammation, insulin resistance, and gut microbiota dysbiosis. However, dietary and lifestyle variables are generally modifiable, behaviorally driven factors rather than involuntary biological variables. Addressing them requires a behavioral or metabolic framework that extends beyond the molecular neurobiology scope of this review.

The decision to restrict the scope was conceptual rather than dismissive. Research on modifiable factors such as nutrition, physical activity, or psychosocial interventions requires distinct methodological frameworks and measurement tools from those used to investigate molecular determinants. Nevertheless, future studies should adopt multidimensional, prospective designs integrating genomic, inflammatory, and neuroimaging data with comprehensive assessment of psychosocial and lifestyle factors. Such an integrative approach is essential for developing more accurate, mechanistically informed predictive models and personalized interventions that better reflect the multifactorial nature of TRD in actual clinical practice.

## Figures and Tables

**Figure 1 ijms-26-11016-f001:**
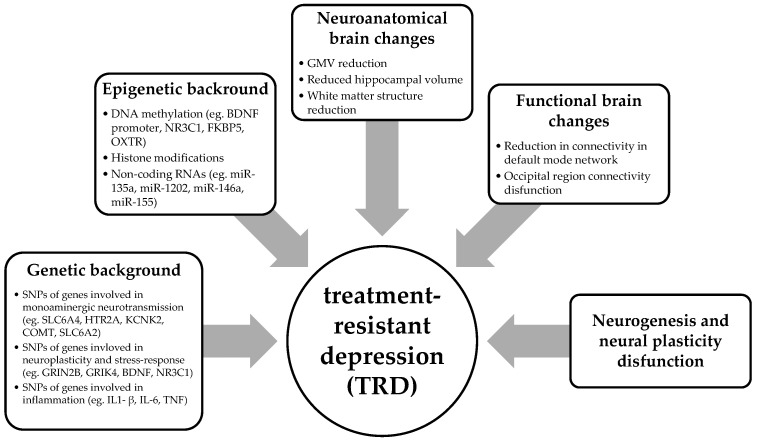
Diagram showing the multidimensional biological background of treatment-resistant depression.

**Table 3 ijms-26-11016-t003:** Functional brain changes related to TRD.

Described Changes	Sample Analyzed	Source
Decreased connectivity between the right caudate and the right middle frontal gyrus and right superior frontal gyrus.	18 TRD patients, 17 patients with first episode of depression	[54]
Reduction in connectivity between parahippocampal gyrus and the left precuneus, left posterior cingulate gyrus, and the left inferior parietal lobe. Reduced connection between inferior parietal lobe and the right caudate. Decreased connectivity between the DMN and other functional networks and brain regions.	17 TRD patients, 17 patients with MDD who responded to treatment, 17 healthy controls	[60]
Decreased functional connectivity between the left and right angular gyrus and the cognitive control network. Decrease in functional connectivity between the anterior and posterior DMN and the visual cortex. Decreased functional connectivity between the motor cortex and other brain regions.	17 TRD patients, 18 patients with MDD who responded to treatment, 18 healthy controls	[61]
Increased local activity in the anterior cingulate cortex and medial frontal gyrus.	18 TRD patients, 17 patients with MDD who responded to treatment, 17 healthy controls	[62]
Increased local in the right thalamus and the supramarginal gyrus, at the edge of the angular gyrus.	16 TRD patients, 16 patients with early-phase non-TRD, 26 healthy controls	[63]
Significant increase in activity in the right middle temporal and the middle cingulate gyri.	22 TRD patients, 22 patients with MDD who responded to treatment, 26 healthy controls	[64]
Decreased interhemispheric functional connectivity in the occipital part of the fusiform gyrus and the calcarine cortex Substantial decreased functional disfunction with other regions in the occipital lobe and with the right inferior temporal gyrus and right insula Increased coherence-based ReHo in the occipital part of the left fusiform gyrus and decreased ALFF values in the lingual gyrus/cuneus	23 TRD patients, 22 patients with MDD who responded to treatment, 19 healthy controls	[58]

## Data Availability

No new data were created or analyzed in this study. Data sharing is not applicable to this article.

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
