# Peer review of "Genetics and Neurobiology of Treatment-Resistant Depression—A Review"

_ijms, 2025, doi:10.3390/ijms262211016_

Round 1
Reviewer 1 Report
Comments and Suggestions for Authors
Summary: The article provides a comprehensive overview of the current state of genetic research in treatment-resistant depression (TRD), appropriately noting the scarcity of TRD-specific studies compared to research on general antidepressant response.
Study Design Specification: The eligibility criteria, while providing a basic framework for study inclusion, lack precision in several key areas. The description conflates observational study designs with interventional trials (e.g., randomized controlled trials), which are methodologically distinct and should be clearly differentiated. Although present in the table, population characteristics are not specified in the methods section, leaving uncertainty about whether animal studies, pediatric cohorts, or adults were included, nor are diagnostic criteria for TRD defined. The focus on “genetics of neurological anomalies” is vague and does not clarify whether it refers exclusively to molecular genetic studies, imaging-genetics, or broader neurobiological markers. Additionally, the absence of defined outcome measures, exclusion criteria, and a publication date range limits reproducibility and transparency. Restricting inclusion to English and Polish publications without justification may also introduce language bias. Greater specificity in these criteria would strengthen the rigor and interpretability of the review.
Genetic Factors: The article provides a valuable discussion of non-coding RNAs, particularly miR-146a and miR-155, which are involved in neuroinflammatory and immune regulatory processes relevant to TRD. However, it omits consideration of genetic variants in inflammatory pathways, such as IL-6, TNF-α, and CRP-related polymorphisms. While the discussion of miR-146a and miR-155 highlights the epigenetic regulation of inflammation, without including inflammatory gene variants, the review misses an opportunity to link genetic predisposition to dysregulated immune signaling. Integrating both genetic and epigenetic contributors to inflammation would provide a more comprehensive understanding of TRD mechanisms and enhance the translational relevance for identifying biomarkers or therapeutic targets.
Discussion: The section presents a solid overview of TRD genetics, clearly outlining limitations of candidate gene studies and GWAS, and highlighting methodological improvements such as sequencing with imputation and pathway-based analyses. Yet, it lacks detail on inflammatory gene variants, and insufficient reporting on sample sizes, overlapping cohorts, and patient-level clinical variables limits interpretability. Integrating these genetic findings with neurobiological and epigenetic data would provide a more complete mechanistic understanding of TRD.
Writing: The manuscript contains several spelling errors and minor typographical mistakes which can reduce clarity and readability. Examples include line 386 (1 an million) and 404 (rea). Addressing these issues through careful proofreading or professional editing would improve the overall quality and professionalism of the manuscript.
Overall, this manuscript addresses an important and evolving area. With revisions to improve clarity, deepen analytical discussion, and enhance integration of genetic, epigenetic, and neurobiological data, the study has the potential to make a meaningful contribution to the literature on TRD and its underlying mechanisms.
Author Response
Below we provide a detailed, point-by-point response to the review.
___
1. “The eligibility criteria, while providing a basic framework for study inclusion, lack precision in several key areas. The description conflates observational study designs with interventional trials (e.g., randomized controlled trials), which are methodologically distinct and should be clearly differentiated. Although present in the table, population characteristics are not specified in the methods section, leaving uncertainty about whether animal studies, pediatric cohorts, or adults were included, nor are diagnostic criteria for TRD defined. The focus on ‘genetics of neurological anomalies’ is vague and does not clarify whether it refers exclusively to molecular genetic studies, imaging-genetics, or broader neurobiological markers. Additionally, the absence of defined outcome measures, exclusion criteria, and a publication date range limits reproducibility and transparency. Restricting inclusion to English and Polish publications without justification may also introduce language bias. Greater specificity in these criteria would strengthen the rigor and interpretability of the review.”
The Materials and Methods section has been revised for clarity, precision, and transparency. The revised text now explicitly states that only adult human participants diagnosed with major depressive disorder according to DSM or ICD criteria and meeting criteria for treatment-resistant depression (defined as lack of response after two adequate antidepressant trials) were included. Excluded were animal studies (animal studies were not included in the primary search, although they were included in the subsequent analysis as an additional source of information), studies on the pediatric population, and on bipolar depression. We clarified that eligible studies must report outcomes directly related to treatment response or resistance (remission, partial response, or non-response). We added a clear time frame — studies published between January 2000 and June 2025 were considered. The limitation to English and Polish publications was justified on the basis of researcher language proficiency and accessibility, and this limitation is now explicitly acknowledged as a potential source of selection bias.
2. “The article provides a valuable discussion of non-coding RNAs, particularly miR-146a and miR-155, which are involved in neuroinflammatory and immune regulatory processes relevant to TRD. However, it omits consideration of genetic variants in inflammatory pathways, such as IL-6, TNF-α, and CRP-related polymorphisms. While the discussion of miR-146a and miR-155 highlights the epigenetic regulation of inflammation, without including inflammatory gene variants, the review misses an opportunity to link genetic predisposition to dysregulated immune signaling. Integrating both genetic and epigenetic contributors to inflammation would provide a more comprehensive understanding of TRD mechanisms and enhance the translational relevance for identifying biomarkers or therapeutic targets.”
We expanded the manuscript to include genetic variants in inflammatory pathways, with the Results section now discussing the SNPs of genes involved in inflammation (including polymorphisms in IL1-β, IL6, TNF-α). Additionally, we tried to emphasize how these variants are associated with antidepressant non-response and neural activation changes, thus linking immune dysregulation with neural mechanisms of treatment resistance
3. “The section presents a solid overview of TRD genetics, clearly outlining limitations of candidate gene studies and GWAS, and highlighting methodological improvements such as sequencing with imputation and pathway-based analyses. Yet, it lacks detail on inflammatory gene variants, and insufficient reporting on sample sizes, overlapping cohorts, and patient-level clinical variables limits interpretability. Integrating these genetic findings with neurobiological and epigenetic data would provide a more complete mechanistic understanding of TRD.”
The Discussion section has been thoroughly rewritten to incorporate inflammatory gene variants and explain their interactions with neural circuits and stress-response pathways, explicitly integrating genetic, epigenetic, and neurobiological findings into a systems-level model of TRD that connects molecular pathways (neuroinflammation, HPA axis dysregulation, neuroplasticity) with neural network alterations.
4. “The manuscript contains several spelling errors and minor typographical mistakes which can reduce clarity and readability. Examples include line 386 (1 an million) and 404 (rea). Addressing these issues through careful proofreading or professional editing would improve the overall quality and professionalism of the manuscript.”
We have carefully proofread and corrected all typographical and grammatical errors throughout the manuscript. Additional corrections were made for consistency in terminology, punctuation, and style. The entire manuscript has been reviewed for language quality to ensure clarity and professional presentation.
___
We are grateful for the detailed comments and thoughtful feedback. We hope that the revised version now addresses all concerns comprehensively.
Reviewer 2 Report
Comments and Suggestions for Authors
The review addresses the genetic and neurobiological mechanisms underlying treatment-resistant depression and covers an important and timely topic. However, the ms could be strengthened in several areas. For instance, important aspects such as the role of inflammation are mentioned only briefly and would benefit from a much deeper examination. Similarly, the involvement of neurogenesis and neural stem cells in the mechanisms of antidepressant action is not addressed and could add valuable insight. Many findings also indicate that physical therapies can be beneficial for treatment-resistant depression and may influence neurogenesis and brain plasticity; including this information would enrich the review. Finally, including a schematic figure linking the described genes and neurobiological mechanisms could greatly improve the clarity and overall readability of the work. On the whole , the review appears superficial and of limited value compared to several similar reviews already published on the topic.
Author Response
Below we provide a detailed, point-by-point response to the review.
___
1. “The review addresses the genetic and neurobiological mechanisms underlying treatment-resistant depression and covers an important and timely topic. However, the ms could be strengthened in several areas.”
In the revised version, we have modified several sections, including a clarification of the Materials and Methods section, an expanded Results section (with the addition of paragraph dedicated to SNPs of genes related to inflammation) and more thorough Discussion section
2. “Important aspects such as the role of inflammation are mentioned only briefly and would benefit from a much deeper examination.”
The revised text discusses the current state of research regarding the potential impact of SNPs of genes of pro-inflammatory cytokines (IL-1β, IL-6, TNF, CRP) in TRD and their role in neurotransmission, HPA-axis dysregulation, and hippocampal atrophy. We cover the role of chronic inflammation on the pathophysiology of TRD and explain why TRD should be considered a disorder correlated with neuroimmune imbalance
3. “Similarly, the involvement of neurogenesis and neural stem cells in the mechanisms of antidepressant action is not addressed and could add valuable insight.”
We have added new content addressing the role of neurogenesis and neural stem-cell activity in TRD, including BDNF signaling and impact of chronic inflammation on neurogenesis.
4. “Many findings also indicate that physical therapies can be beneficial for treatment-resistant depression and may influence neurogenesis and brain plasticity; including this information would enrich the review.”
We have incorporated a paragraph in the Discussion summarizing evidence for physical and neuromodulatory therapies, including ECT and TMS therapy
5. “Finally, including a schematic figure linking the described genes and neurobiological mechanisms could greatly improve the clarity and overall readability of the work.”
We have added a new Figure 1 entitled “Diagram showing the multidimensional biological background of treatment-resistant depression.”
___
We hope that the revisions address the reviewer’s concerns and therefore improve the manuscript’s scientific value and clarity. We are grateful for the thoughtful feedback and the opportunity to strengthen our work.
Round 2
Reviewer 1 Report
Comments and Suggestions for Authors
The authors have thoroughly addressed all my concerns, and the manuscript is significantly improved. I recommend accepting the paper.
Author Response
Thank you once again for an in-depth review and many insightful comments.
Reviewer 2 Report
Comments and Suggestions for Authors
The review appears overall superficial and lacks originality compared to the existing literature. It fails to provide in-depth discussion on two central topics that are now extensively studied and highly relevant to the field: neurogenesis/neural plasticity and nutritional factors (with particular reference to the effects of simple sugars). Moreover, the authors do not develop the important issue concerning the pharmacological effects of antidepressant drugs hippocampal neurogenesis, and behavioral outcomes.
In conclusion, I think the ms does not deserve publication
Author Response
We would like to once again thank for the honest review and feedback. We further modified the manuscript to address the issues raised.
A new section “Neurogenesis and Neural Plasticity Dysfunction” has been added. It further discusses structural correlates of neurogenesis (including hippocampal alterations, described in more detail in earlier sections of the manuscript), the role of BDNF and VEGF in impaired neuroplasticity observed in TRD, as well as the influence of chronic stress and inflammation on neurogenesis and synaptic remodeling. These additions directly address the Reviewer’s observation regarding the need for a more in-depth discussion of neural plasticity and neurogenesis. As adding such information required additional research, the Materials and Methods section has been updated accordingly
Additionally, the Discussion section has been improved, with more considerations on the issue of chronic inflammation and how it impairs neurogenesis and contributes to antidepressant resistance, thus presenting a more integrative perspective on TRD pathophysiology. The revised Discussion also elaborates on pharmacological effects of antidepressants on hippocampal neurogenesis, including the interaction between monoaminergic, glutamatergic, and neurotrophic pathways. Now mentioned is also the reasoning behind ketamine and psylocybin efficiency in TRD.
We acknowledge the Reviewer’s mention of nutritional factors (including the effects of simple sugars). However, dietary and nutritional aspects were intentionally excluded from this review, as the focus of the paper is strictly on biological and molecular determinants of treatment resistance. Nutritional factors are behaviorally driven and modifiable lifestyle elements, not involuntary biological variables, and were therefore considered outside the scope of a molecular neurobiology review. Including them would have required a behavioral or metabolic framework rather than a neurobiological one.
Other general improvements include update of the abstract as well as an expanded list of keywords.
In summary, the revised manuscript now provides a much deeper view of TRD pathophysiology in terms of its neurobiology.